# Neuroinvasive and neurovirulent potential of SARS-CoV-2 in the acute and post-acute phase of intranasally inoculated ferrets

**Feline F. W. Benavides, Edwin J. B. Veldhuis Kroeze, Lonneke Leijten, Katharina S. Schmitz, Peter van Run, Thijs Kuiken, Rory D. de Vries, Lisa Bauer[☉], Debby van Riel**[iD][☉]*

Department of Viroscience, Erasmus MC, Rotterdam, The Netherlands

[☉] These authors contributed equally to this work.

* d.vanriel@erasmusmc.nl

## Abstract

Severe acute respiratory syndrome corona virus 2 (SARS-CoV-2) can cause systemic disease, including neurological complications, even after mild respiratory disease. Previous studies have shown that SARS-CoV-2 infection can induce neurovirulence through microglial activation in the brains of patients and experimentally inoculated animals, which are models representative for moderate to severe respiratory disease. Here, we aimed to investigate the neuroinvasive and neurovirulent potential of SARS-CoV-2 in intranasally inoculated ferrets, a model for subclinical to mild respiratory disease. The presence of viral RNA, histological lesions, virus-infected cells, and the number and surface area of microglia and astrocytes were investigated. Viral RNA was detected in various respiratory tissue samples by qPCR at 7 days post inoculation (dpi). Virus antigen was detected in the nasal turbinates of ferrets sacrificed at 7 dpi and was associated with inflammation. Viral RNA was detected in the brains of ferrets sacrificed 7 dpi, but *in situ* hybridization nor immunohistochemistry did confirm evidence for viral RNA or antigen in the brain. Histopathological analysis of the brains showed no evidence for an influx of inflammatory cells. Despite this, we observed an increased number of Alzheimer type II astrocytes in the hindbrains of SARS-CoV-2 inoculated ferrets. Additionally, we detected increased microglial activation in the olfactory bulb and hippocampus, and a decrease in the astrocytic activation status in the white matter and hippocampus of SARS-CoV-2 inoculated ferrets. In conclusion, although SARS-CoV-2 has limited neuroinvasive potential in this model for subclinical to mild respiratory disease, there is evidence for neurovirulent potential. This study highlights the value of this ferret model to study the neuropathogenecity of SARS-CoV-2 and reveals that a mild SARS-CoV-2 infection can affect both microglia and astrocytes in different parts of the brain.

## Introduction

Severe acute respiratory syndrome coronavirus 2 (SARS-CoV-2) infection causes respiratory disease. Additionally, a broad variety of neurological symptoms has been reported in the acute and post-acute phases. These neurological complications can occur after severe, mild or

**Data availability statement:** All relevant data are within the paper and its Supporting Information files.

**Funding:** This work was funded in part by a fellowship from the Netherlands Organization for Scientific Research (VIDI contract 91718308). Ferret experiments were previously published and supported by funding from the National Institutes of Health (AI146980, AI121349, NS091263, AI114736, HHSN272201400008C).

**Competing interests:** The authors have declared that no competing interests exist.

even subclinical respiratory disease and comprise a variety of symptoms. In the acute phase, symptoms including brain fog, headache and memory loss have been observed and several of these symptoms can also persist in the post-acute phase and are part of long-COVID [1,2]. Yet, the pathogenesis of neurological disease resulting from SARS-CoV-2 infection is not fully understood.

SARS-CoV-2 has the ability for neuroinvasion, meaning that the virus can enter the peripheral or central nervous system (PNS, CNS) [3]. Several studies have shown that SARS-CoV-2 can enter the CNS through axonal transport via several cranial nerves (the olfactory and trigeminal nerves, among others) or via the hematogenous route [3]. One important route is via the olfactory mucosa (located in the nasal cavity) where replication of SARS-CoV-2 can result in virus spread through the cribriform plate into the olfactory bulb of the brain. In the olfactory mucosa of coronavirus disease 2019 (COVID-19) patients, SARS-CoV-2 replicates predominantly in sustentacular cells and rarely in olfactory receptor neurons based on the expression of viral antigen or viral RNA [4,5]. In tissues obtained from human autopsies, viral RNA or virus antigen was detected in the olfactory bulb [5–8]; however, other studies failed to confirm this [8,9]. Virus antigen and viral RNA have been detected in neurons, astrocytes and oligodendrocytes in other areas of the human CNS, like the frontal lobe, hypothalamus, basal ganglia, cerebellum and spinal cord [9–13]. Virus antigen and viral RNA were detected in the olfactory bulb of experimentally inoculated cynomolgus macaques, ferrets, mice and hamsters [14–18], suggesting neuroinvasion via the olfactory nerve. *In vitro*, SARS-CoV-2 infects neurons in human brain organoids, human pluripotent stem cell (hPSC)-derived dopaminergic neurons and hPSC-derived cortical neurons and astrocytes [18–22]. Altogether, these studies show that SARS-CoV-2 has both a neuroinvasive and neurotropic potential, although the frequency of neuroinvasion is not well understood.

Neurovirulence of SARS-CoV-2 infection is its ability to cause lesions in the CNS leading to clinical disease, independent of direct infection of CNS cells [3]. Several mechanisms, such as microgliosis and astrogliosis, are thought to contribute to the neurovirulence of SARS-CoV-2 infection. The activation of microglia (microgliosis) and astrocytes (astrogliosis) is characterized by morphological changes of these cell types as well as their influx into affected areas. In deceased COVID-19 patients, more extensively ramified microglia were detected in the olfactory bulb, white matter and medulla oblongata [11,23,24], as well as an increased influx of microglia into the olfactory bulb, meninges, brainstem and cerebellum [7,11,25–27]. Astrogliosis was observed in grey and white matter, in the olfactory bulb, frontal cortex, medulla oblongata, cerebellum and basal ganglia of COVID-19 patients [7,11,24]. Transcriptomic analysis comparing post-mortem brain tissues of control and COVID-19 patients revealed a significant enrichment of astrocyte-associated genes in the amygdala potentially indicating activation of astrocytes [6]. Similar to humans, microglial activation in the olfactory bulb, hippocampus, cortex and medulla oblongata was also observed in experimentally inoculated mice and hamsters [17,23,24,28]. Furthermore, enlargement of astrocytes was observed in the olfactory bulb and hippocampus of hamsters [28], while another study failed to report an influx of astrocytes in the olfactory bulb, cortex, medulla oblongata and hippocampus of SARS-CoV-2 inoculated hamsters [24].

Altogether, *in vivo* studies have shown that SARS-CoV-2 has a neuroinvasive and neurovirulent potential in humans and animal models for moderate to severe respiratory disease, like mouse and hamster. Whether SARS-CoV-2 infection can also induce neurovirulence during a subclinical or mild infection is currently unknown. Therefore, we investigated the neuroinvasive and neurovirulent potential of SARS-CoV-2 in the ferret (*Mustela putorius furo*) that develop subclinical to mild respiratory disease. Here, we characterized the neuroinvasive and neurovirulent potential of the ancestral SARS-CoV-2 in intranasally inoculated ferrets, and

investigated the presence of histological lesions, viral RNA, and virus-infected cells, as well as numbers and surface areas of microglia and astrocytes in the ferret brains. In this study, ferrets have been examined at the acute and post-acute phase of SARS-CoV-2 infection. We determined acute phase up to 10 dpi, when the virus has not yet been cleared, and the post-acute phase after 10 dpi, which can potentially provide insights into long-COVID.

## Materials and methods

### Animal experiments and tissues collection

Tissues from intranasally inoculated SARS-CoV-2 ferrets from a previous study were included. These experiments were conducted in compliance with the Dutch legislation [29] for the protection of animals used for scientific purposes (2014, implementing EU Directive 2010/63) and other relevant regulations. All animal experiments were performed under BSL-3 conditions at Erasmus MC. In short, influenza A virus, SARS-CoV-2 and Aleutian Disease Virus seronegative male and female ferrets (Triple F Farms, PA, USA; between 1-2 years old) were intranasally inoculated with $10^5$ TCID$_{50}$/ml SARS-CoV-2 D614G (isolate BetaCoV/Munich/BavPat1/2020). Ferrets were housed in groups of 3-6 animals, received standard feed on a daily basis and had access to water ad libitum. Cages contained several sources of environmental enrichment and hiding or sleeping places. Four ferrets were bled via a heart puncture 7 dpi, and three ferrets at 21 dpi. All animal handling, including euthanasia, was performed under anaesthesia with a mixture of ketamine/medetomidine (10mg/kg and 0.05mg/kg, respectively) antagonized by atipamezole (0.25 mg/kg). Animal welfare was monitored on a daily basis. Human endpoints included (1) when animals stopped eating or drinking; (2) >20% of weight loss; (3) moderate increase of breathing frequency; (4) moderate impaired behaviour or movement; (5) moderate clinical disease (abnormal posture, ruffled fur, signs of dehydration) and activity score <2 according to the Reuman scoring system. Only female ferrets that did not receive any treatment were included in this study. Three age-matched and sex-matched non-inoculated ferrets were used as negative control for comparison of histological scoring. The brain, trigeminal nerve, tip of the nose, nasal septum, and the nasal turbinates, which contained olfactory and respiratory mucosa were sampled for histopathological analysis. Tissues were fixed in 10% formalin for two weeks. Transverse sections of the fixed brain were made at 0.5 cm intervals to evaluate spatial differences in histopathological changes and subsequently embedded in paraffin (Fig 1A).

### Immunohistochemistry and histopathology

Three-μm-thick formalin-fixed paraffin-embedded (FFPE) tissue sections were deparaffinized, rehydrated and stained with haematoxylin (Klinipath) and eosin (QPath; H&E) to assess histopathological changes. For immunohistochemistry (IHC), consecutive tissue sections were deparaffinized, rehydrated and pre-treated for antigen retrieval. Pre-treatment of slides consisted of boiling for 15 minutes in citric acid buffer (pH 6.0) for the detection of SARS-CoV-2 nucleoprotein (NP) and GFAP (glial fibrillary acidic protein; astrocytes); or boiling for 15 minutes in TRIS-EDTA buffer (pH 9.0) for the detection of IBA1 (ionized calcium binding adaptor molecule 1; microglia). After antigen retrieval, slides were incubated with 3% hydrogen peroxidase (Sigma-Aldrich) for 10 minutes to block endogenous peroxidases. After blocking, slides were washed with phosphate-buffered saline (PBS), and PBS/0.05% Tween 20 (Merck). Sections for SARS-CoV-2 IHC were additionally blocked with 10% goat serum (X0907, DAKO) for 30 minutes at room temperature (RT). Slides were incubated with a rabbit polyclonal antibody against SARS-CoV/SARS-CoV-2-nucleoprotein (40143-T62, Sino Biological, 1:1000), rabbit anti-IBA1 (019-19741, Wako Pure Chemical Corporation; 2.5 μg/

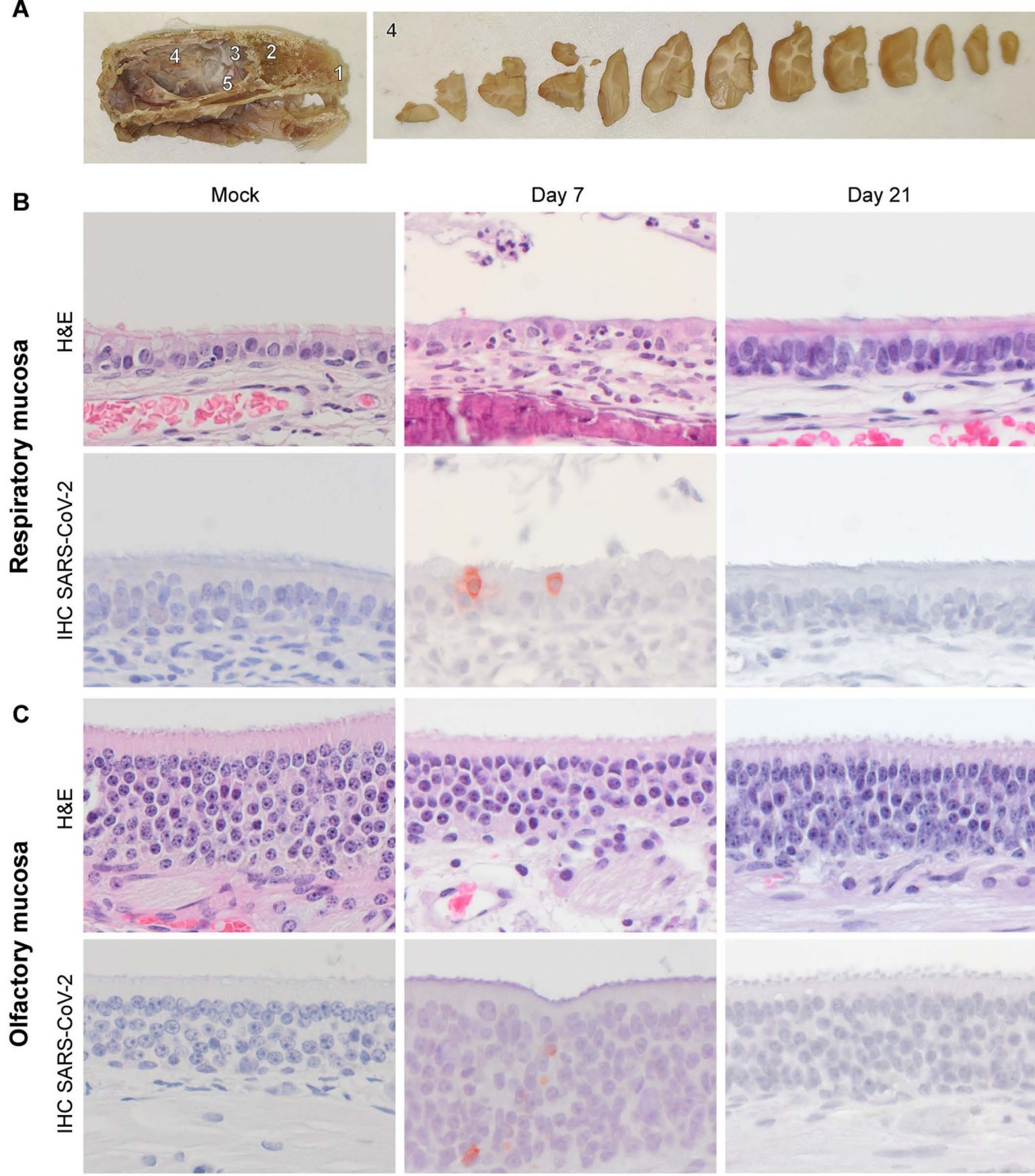

**Fig 1. Histology and presence of SARS-CoV-2 antigen expression in the respiratory and olfactory mucosa of SARS-CoV-2 inoculated ferrets.** Intra-nasally inoculated ferrets (D614G SARS-CoV-2; $10^5$ TCID$_{50}$) were sacrificed at day 7 or day 21 post inoculation. (A) Left panel: Cross section of a ferret head with the different anatomical locations: 1 = tip of the nose; 2 = nasal cavity containing respiratory and olfactory mucosa; 3 = olfactory bulb; and 4 = the brain within the cranial cavity; 5 = trigeminal nerve. Right panel, transverse sections of the fixed brain were made at 0.5 cm intervals to evaluate

spatial differences in histopathological changes. (B-C) Hematoxylin and eosin (H&E) staining and immunohistochemistry (IHC) staining for SARS-CoV-2 nucleoprotein of the respiratory mucosa (B) and olfactory mucosa (C) of SARS-CoV-2 inoculated ferrets. In the respiratory mucosa, individual columnar ciliated epithelial cells express cytoplasmic SARS-CoV-2 antigen at 7 dpi (B, lower row) colocalized with infiltration of neutrophils, and intra-luminal presence of cellular debris also containing few neutrophils (B, upper row). In the olfactory mucosa, individual cells contained virus antigen.

ml) or mouse anti-GFAP (clone 4A11, BD556327, Biosciences, 20 µg/ml) in PBS/0,1% BSA (Aurion) for 1h at RT. For each staining on each tissue an isotype control was included as a negative control, either a rabbit IgG isotype control (AB-105-C, R&D, 1:200; 2.5 µg/ml), or mouse IgG2b isotype control (MAB0041, R&D, 5 µg/ml). Tissue sections from negative control ferrets were included as negative tissue controls for all stainings. A lung section from an experimentally inoculated hamster with SARS-CoV-2 was used as positive control for the NP SARS-CoV-2 IHC. After washing, sections were incubated with peroxidase labelled goat-anti-Rabbit IgG (P0448, DAKO, 1:100) or goat-anti-mouse IgG (PO447, Southern Biotech, 1:100) in PBS/0,1% BSA for 1 hour at RT. Peroxidase activity was revealed by incubating slides in 3-amino9-ethylcarbazole (AEC; Tokyo Chemical Industry) in N,N-dimethylformamide (DMF; Sigma-Aldrich) for 10 minutes, resulting in a bright red precipitate, followed by counterstaining with haematoxylin (Sigma-Aldrich).

## RNA isolation and qPCR

RNA was extracted from paraffin sections of all available tissues using the RNeasy FFPE kit (Qiagen) according to the manufacturer's protocol. In short, sections of 5 µM were collected in Eppendorf vials and deparaffinized using xylene. The sample was lysed with proteinase K digestion followed by heat and DNAse treatment. Ethanol and RBC buffer from the kit was added to bind total RNA to the spin columns. Samples were washed and eluted twice in RNAse free water. qPCR was performed targeting E-gene of SARS-CoV-2 as previously reported [30]. For the qPCR, the maximum RNA input was used, which corresponds to 100 ng input. The housekeeping gene hypoxanthine phosphoribosyltransferase 1 (HPRT) was included as control; its sequence can be found in [31]. All reactions were run on a 7500 Real Time PCR Cycler (Applied Biosystems) in technical duplicates. A Ct-value < 45 was considered positive. Values are depicted ± SD. For the brain, all separate tissue sections were tested individually, but the average is depicted ± SD.

## SARS-CoV-2 in situ hybridization

BaseScope™ RNA probes were designed by Bio-Techne Ltd (Abingdon, UK) for SARS-CoV-2 BA-V-CoV-Wuhan-Nucleocapsid-3zz-st (846661). In situ hybridization (ISH) was performed on FFPE consecutive sections using BaseScope™ Reagent Kit v2–RED (323900) as described by the manufacturer. SARS-CoV-2 nucleocapsid RNA molecules were visualized as red chromogenic dots.

## Microscopy

H&E sections of all tissues were evaluated for histopathological changes with an Olympus BX51 light microscope by a certified veterinary pathologist. A 40 × digital scan was made of H&E slides of the hindbrain (area surrounding the pons and cerebellum) for counting Alzheimer type II astrocytes using a Hamamatsu Nanozoomer 2.0 HT digital slide scanner with the accompanying software (NanoZoomer Digital Pathology and NDP.scan and NDP.view, Hamamatsu, Higashiku, Hamamatsu City, Japan). For microglial and astrocytic evaluation, a 10 × air objective (Olympus) was used to select a region of interest with an Olympus BX51 microscope. The regions of interest for microglial and astrocytic evaluation in this study were

determined as the glomerular layer and the granular cell layer of the olfactory bulb, the white matter tract (corpus callosum), grey matter (posterior parietal cortex) and the hippocampus (CA1 region). We chose these areas, like the corpus callosum, to consistently sample in the same brain area, and therefore a ferret brain atlas was used to determine the correct area for imaging [32]. The slides were blinded before the images of the selected regions of interest were taken. Images were taken at either a 200× magnification (20× air objective; Olympus) or 400× magnification (40× air objective; Olympus) with CellSense software.

### Quantitative analysis of IBA1+ cells, GFAP+ cells and Alzheimer type II astrocytes

At least two sections per brain area were selected per ferret, and per section at least three images were taken of the region of interest in the brain. The number of IBA1+ (microglia) and GFAP+ (astrocytes) cells per high power field (400× magnification) was determined by manual counting by two observers blinded to the infection status. The averages ± SD of the IBA1+ or GFAP+ cells per ferret were plotted. As a measure for activation of astrocytes and microglia, the surface areas of IBA1+ and GFAP+ cells was determined using the pixel classifier function in QuPath 0.4.4 [33] with a Gaussian filter. The threshold values to determine the positive and negative surface areas can be found in Table 1. The averages ± SD of the IBA1+ and GFAP+ cells surface areas per ferret were plotted. Additionally, the surface area per IBA1+ or GFAP+ cell was plotted ± SD.

Alzheimer type II astrocytes were counted manually from a 40× digital scan of a H&E slide in Qupath 0.4.4 if the cell matched to the description referenced before [34,35]. Alzheimer type II astrocytes are characterized by a large, pale nucleus, with a rim of chromatin and a small conspicuous nucleolus. The total brain area in the slide was calculated in $\mu m^2$ in Qupath 0.4.4 [33], and the number of Alzheimer type II astrocytes was normalized to number of cells per $mm^2$ brain area and plotted ± SD.

### Statistical analysis

Statistical differences between experimental groups were determined by using a one-way analysis of variance (ANOVA) with a Dunnett's *posthoc* test using GraphPad Prism version 10.1.2. Additional information about the data and statistics is provided in the figure legends. P values of ≤ 0.05 were considered statistically significant. Figures were prepared with Adobe Photoshop 22.1.1 and Adobe Illustrator 25.1.0.

## Results

### Assessment of ferret respiratory tissues

To evaluate the cell tropism in the respiratory tract of D614G SARS-CoV-2 intranasally inoculated ferrets were sacrificed at 7 or 21 dpi and samples of the tip of the nose, nasal septum,

Table 1. The chosen threshold values used in QuPath to determine the IBA1+ and GFAP+ surface area.

| Region of interest | IBA1+ surface area | GFAP+ surface area |
|---|---|---|
| Glomerular layer (olfactory bulb) | 0.12 | N.P. |
| Granular cell layer (olfactory bulb) | 0.12 | N.P. |
| White matter (corpus callosum) | 0.23 | 0.20 |
| Grey matter (posterior parietal cortex) | 0.18 | 0.21 |
| Hippocampus (CA1 region) | 0.17 | 0.17 |

N.P. = not performed, IBA1 = ionized calcium binding adaptor molecule 1, GFAP = glial fibrillary acidic protein.

nasal turbinates containing olfactory and respiratory mucosa were analysed. As reported earlier, SARS-CoV-2 RNA was detected by qPCR in swabs of the throat and nose of these ferrets at 7 dpi [29]. Additionally, infectious virus was detected in throat swabs of these ferrets at 3 dpi [29]. In our analyses on the FFPE tissues, at 7 dpi, viral RNA was detected in tissues of the nasal septum and nasal turbinates in all ferrets (Table 2). At 21 dpi, viral RNA was detected in the tip on the nose (1 out of 3 ferrets) and nasal turbinates (2 out of 3 ferrets; Table 2). ISH verified the presence of the SARS-CoV-2 viral RNA in all qPCR-positive respiratory tissue sections. Next, to determine the location and cellular tropism of SARS-CoV-2 in the ferret respiratory tract, IHC for SARS-CoV-2 NP was performed. Virus antigen could not be detected in the tip of the nose or nasal septum from ferrets sacrificed at 7 and 21 dpi. Virus antigen was detected at 7 dpi in both respiratory and olfactory mucosa of the nasal turbinates, but was more pronounced in the ciliated respiratory epithelial cells (Fig 1B). The presence of virus antigen was associated with inflammation, characterized by slight to mild infiltration of neutrophils within the mucosal lamina propria and within the epithelial lining in the respiratory mucosa and in a lesser extent in the olfactory mucosa. Additionally, necrotic sloughed epithelial debris admixed with few neutrophils was present within nasal passages along the inflamed mucosae in one ferret (Fig 1B). Virus antigen or histopathological changes of significance were not detected in the respiratory tissues of ferrets sacrificed at 21 dpi, although sporadic numbers of neutrophils were still present within the lamina propria and within the epithelial lining of the respiratory mucosa of two ferrets (Fig 1C).

**Table 2. Ct-values of viral RNA detected in various tissues of SARS-CoV-2 inoculated ferrets.**

| | | | Ferret | | | | | | |
|---|---|---|---|---|---|---|---|---|---|
| | Days post inoculation | Target | #1 | #2 | #3 | #4 | #5 | #6 | #7 |
| **Nasal turbinates** | 7 | S2 | 32.2 | 28.3 | 27.0 | 32.8 | | | |
| | | HPRT | 37.9 | 37.2 | 34.5 | 34.1 | | | |
| | 21 | S2 | | | | | 37.3 | N.D. | 32.8 |
| | | HPRT | | | | | 39.0 | 34.9 | 34.8 |
| **Tip of the nose** | 7 | S2 | N.D. | N.D. | N.D. | N.D. | | | |
| | | HPRT | 36.1 | 35.5 | 37.2 | 34.7 | | | |
| | 21 | S2 | | | | | N.D. | 39.1 | N.A. |
| | | HPRT | | | | | 36.8 | 36.4 | N.A. |
| **Nasal Septum** | 7 | S2 | 36.6 | 35.9 | 34.1 | 33.0 | | | |
| | | HPRT | 35.9 | 38.2 | 37.2 | 34.2 | | | |
| | 21 | S2 | | | | | N.D. | N.D. | N.D. |
| | | HPRT | | | | | 39.5 | 35.8 | 38.9 |
| **Trigeminal nerve** | 7 | S2 | N.D. | N.D. | N.D. | N.A. | | | |
| | | HPRT | 37.2 | 37.7 | 37.1 | N.A. | | | |
| | 21 | S2 | | | | | N.D. | N.A. | N.D. |
| | | HPRT | | | | | 38.7 | N.A. | 42.0 |
| **Brain (pooled)** | 7 | S2 | 41.5 | 39.0 | 38.6 | 38.5 | | | |
| | | HPRT | 37.5 | 36.6 | 35.4 | 35.3 | | | |
| | 21 | S2 | | | | | N.D. | N.D. | N.D. |
| | | HPRT | | | | | 37.1 | 35.4 | 36.1 |

S2 = SARS-CoV-2; HPRT = hypoxanthine phosphoribosyltransferase 1 (housekeeping gene); N.D. = not detected, Ct-value $\geq$ 45; N.A. = not available; Nasal turbinates contain both respiratory and olfactory mucosa; Brain (pooled) is pooled data from all positive qPCR brain tissue sections.

## Assessment of the neuroinvasive potential of SARS-CoV-2 in ferrets

Next, we investigated the neuroinvasive potential of SARS-CoV-2 in ferrets, by testing for viral RNA by qPCR and virus antigen expression by IHC. If viral RNA was detected, we verified this by ISH. Viral RNA was detected by qPCR in inoculated ferrets sacrificed at 7 dpi in various parts of the brain (pooled results of various qPCR positive brain tissues sections in Table 2). However, virus antigen was not detected by IHC in any ferrets and ISH did not confirm the presence of viral RNA in qPCR-positive brain tissues.

## Assessment of histopathological changes in the brain

All brain areas were screened for histopathological abnormalities. We did not detect infiltration of any inflammatory cells, and no gliosis or necrosis in the brain tissues of any ferret. Alzheimer type II astrocytes were detected in the area of the pons and the cerebellum (example in Fig 2A). We observed significantly more Alzheimer type II astrocytes in SARS-CoV-2 inoculated ferrets sacrificed at 7 and 21 dpi compared to control (Fig 2B). The number of Alzheimer type II astrocytes in ferrets sacrificed at 21 dpi were significantly lower compared to ferrets sacrificed at 7 dpi (Fig 2B).

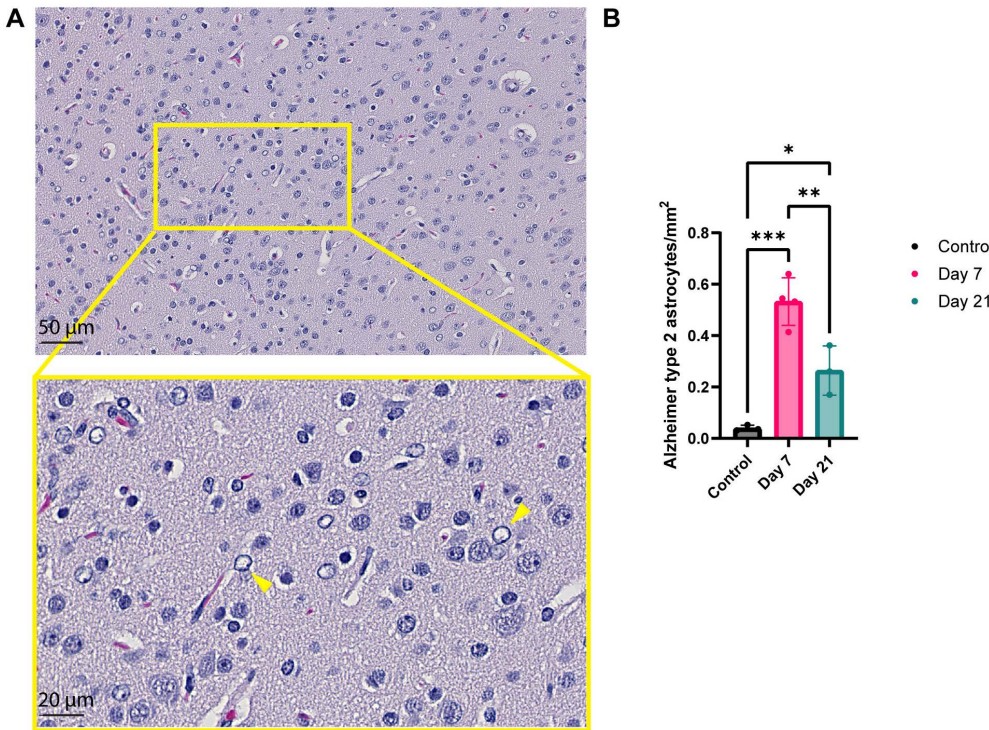

**Fig 2. Detection of Alzheimer type II astrocytes in the area of the pons and cerebellum of inoculated ferrets.** ( A) Alzheimer type II astrocytes (example with yellow arrows), which were characterized by having a large, pale nucleus with a rim of marginated chromatin and small conspicuous nucleolus. (B) The number of Alzheimer type II astrocytes were counted in a 40x digital scan from a brain slice and normalized to number of cells per mm². Statistical significance was calculated with a one-way analysis of variance (ANOVA) with a Dunnett's *posthoc* test. Averaged values of three or four individual ferrets per group were compared to values of three control ferrets. Data is shown as mean ± SD. Asterisks indicate statistical significance (*P < 0.05, **P < 0.01, ***P < 0.001).

## Assessment of neurovirulence in the olfactory bulb

Expression of IBA1 (microglia) and GFAP (astrocytes) was determined in the brain of SARS-CoV-2 inoculated ferrets by IHC to characterize neurovirulence. IBA1 expression in the glomerular layer of olfactory bulbs appeared to be modestly increased by qualitative visual examination in ferrets sacrificed at 7 dpi and 21 dpi compared to the control ferrets (Fig 3A). Quantitative analysis revealed no significant differences in the number of IBA1+ cells between control and infected ferrets in both the glomerular and granular cell layers of the olfactory bulb (Fig 3B, C). However, there was a significant increase in the IBA1+ surface area per high power field (HPF) in the glomerular layer of the olfactory bulb in ferrets sacrificed at 7 and 21 dpi compared to control ferrets (Fig 3B). Analysis of the surface area normalized per IBA1+ cell did not differ significantly between infected and control ferrets in the glomerular layer and granular cell layer of the olfactory bulb (Fig 3B, C). The GFAP expression did not appear to be different between control and infected ferrets (Fig 3D). Quantification of the number and surface area of GFAP+ cells was not feasible in the olfactory bulb, because GFAP expression was to widespread which did not allow us to distinguish single cells, which is required for quantification.

## Assessment of neurovirulence in white and grey matter of cerebral cortex and the hippocampus

IBA1 and GFAP IHC was performed to investigate the activation status of microglia and astrocytes, respectively, in the white and grey matter of the cerebral cortex (corpus callosum and posterior parietal cortex, respectively) and the hippocampus (CA1 region). Again, quantitative analyses were performed per brain area: counting the number of IBA1+ or GFAP+ cells and measuring the IBA1+ or GFAP+ surface area per HPF (Fig 4). In the white matter, no statistical significant differences in the number of IBA1+ cells, surface area or surface area per IBA1+ cell were observed (Fig 4A), while significantly less GFAP+ surface area was detected at 7 and 21 dpi (Fig 4B). In the grey matter, no statistical significant differences were observed in the number of IBA1+ and GFAP+ and their surface area (Fig 4C, D). In the hippocampus, a statistical significant increase in IBA1+ surface area was detected (Fig 4E) in inoculated ferrets sacrificed at 7 and 21 dpi compared to control ferrets, while a decrease in GFAP+ surface was seen in ferrets sacrificed at 7 dpi compared to control ferrets (Fig 4F).

## Discussion

This study demonstrates that SARS-CoV-2 has limited neuroinvasive potential via the olfactory route in this ferret model, although neuroinvasion via the hematogenous route or neuroinvasion via cranial nerves at earlier timepoints cannot be excluded. We observed SARS-CoV-2 replication in the respiratory and olfactory mucosa confirming other studies [4,5,14,16,17,29], as well as the detection of viral RNA in the CNS [16]. The ability of SARS-CoV-2 to replicate in the olfactory mucosa in experimentally inoculated ferrets that develop mild respiratory disease, increases the likelihood that virus can spread to the brain via the olfactory nerve as observed in influenza A virus-infected ferrets [36] or in SARS-CoV-2 infected hamsters [17].

Subtle changes were observed in the brains of SARS-CoV-2 inoculated ferrets, despite the lack of strong evidence for neuroinvasion. These changes included microgliosis and a decrease in astrocytic activation, as well as, an increase in Alzheimer type II astrocytes in the acute and post-acute phase. The neurovirulence is likely very complex and multifactorial. Microglia and astrocytes can be reactive upon direct infection or stimuli in their environment such as the presence of viruses or other pathogens in surrounding cells [17,37–42]. Another way

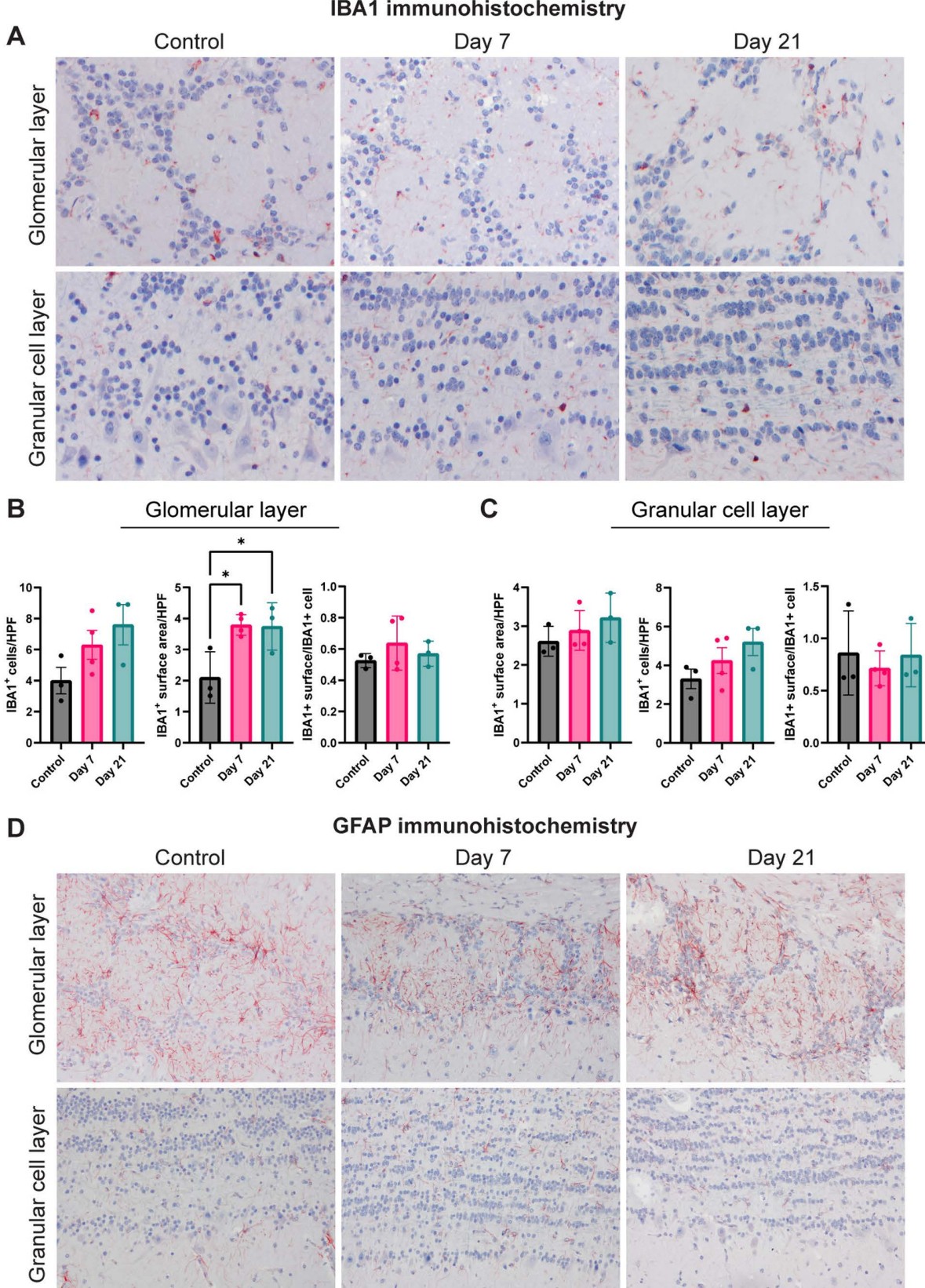

**Fig 3. Activation status of microglia and astrocytes in the olfactory bulb of SARS-CoV-2 inoculated ferrets.** (A) Detection of microglial (IBA1+) cells in the glomerular and granular cell layer of the olfactory bulbs of SARS-CoV-2 inoculated ferrets. (B-C) The number of

IBA1+ cells and IBA1+ surface area in the glomerular layer (B) and granular cell layer (C) of the olfactory bulb. Statistical significance was calculated with a one-way analysis of variance (ANOVA) with a Dunnett's *posthoc* test. Averaged values of three or four individual ferrets per infection group were compared to values of three control ferrets. Data shown as mean ± SD. Asterisks indicate statistical significance (*P < 0.05). (D) Detection of astrocytes (GFAP+) cells in the glomerular and granular cell layer of olfactory bulb of inoculated ferrets. Abbreviations: HPF = high power field, IBA1 = ionized calcium binding adaptor molecule 1, GFAP = glial fibrillary acidic protein.

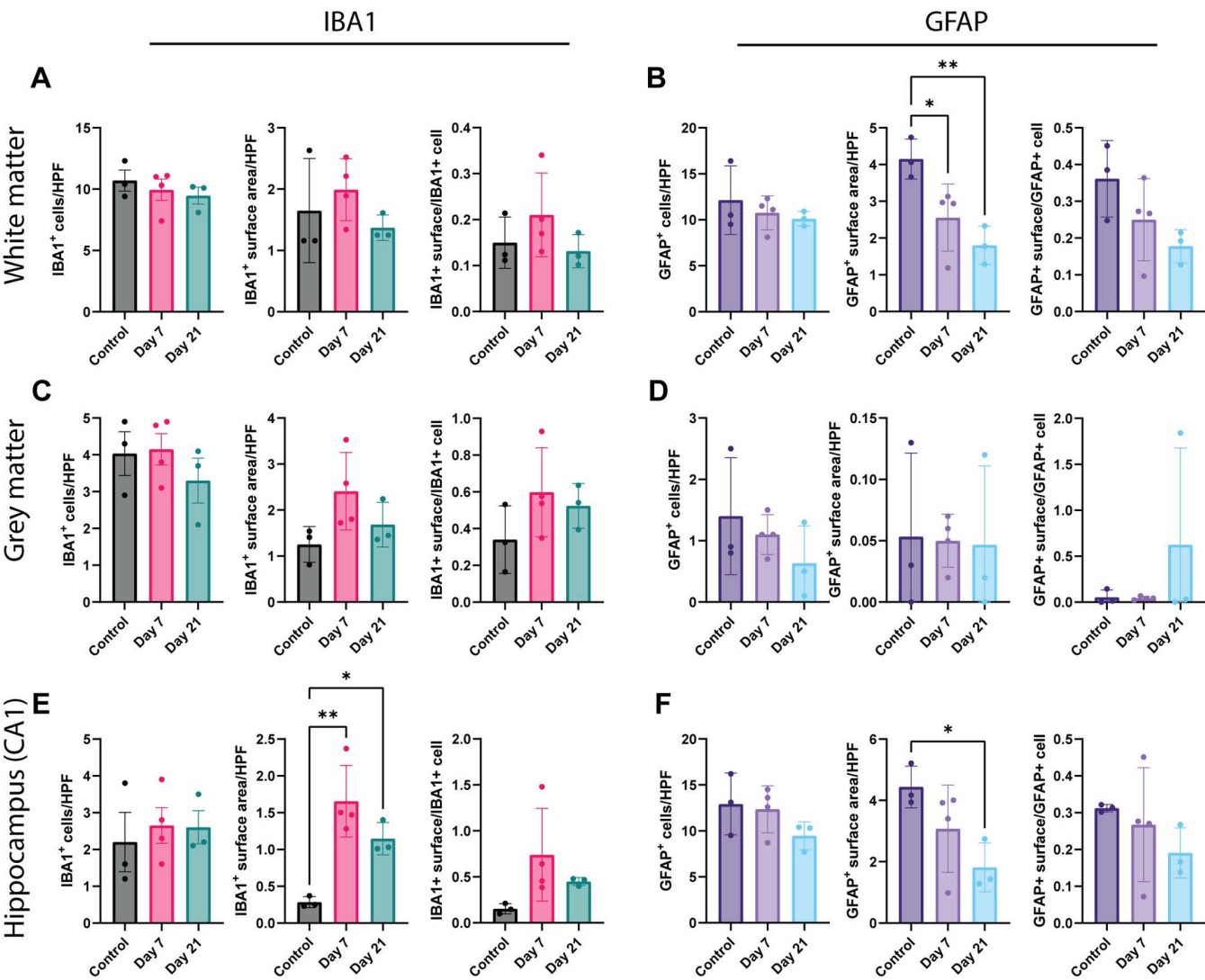

**Fig 4. The activation status of microglia and astrocytes in the white matter, grey matter and hippocampus of SARS-CoV-2 inoculated ferrets at 7 and 21 dpi** (A) Number of IBA1+ cells, the IBA1+ surface area and the surface area per cell in the white matter. (B) Number of GFAP+ cells, the GFAP+ surface area and the surface area per cell in the white matter. (C) Number of IBA1+ cells, the IBA1+ surface area and the surface area per cell in the grey matter. (D) Number of GFAP+ cells, the GFAP+ surface area and the surface area per cell in the grey matter. (E) Number of IBA1+ cells, the IBA1+ surface area and the surface area per cell in the hippocampus (CA1 region). (F) Number of GFAP+ cells, the GFAP+ surface area and the surface area per cell in the hippocampus (CA1 region). Statistical significance was calculated with a one-way analysis of variance (ANOVA) with a Dunnett's *posthoc* test. Averaged values of three or four individual ferrets per infection group were compared to values of three control ferrets. Data shown as mean ± SD. Asterisks indicate statistical significance (*P < 0.05, **P < 0.01). Abbreviations: HPF = high power field, IBA1 = ionized calcium binding adaptor molecule 1, GFAP = glial fibrillary acidic protein.

for increased activation is through indirect stimuli like proinflammatory cytokines caused by a systemic response to for example viral infection [40,41], or a combination of all above. Increased microglial activation upon SARS-CoV-2 infection was observed in patients and experimentally inoculated mice and hamsters, which are models for severe respiratory disease [7,11,17,23–28]. Microglia play a crucial role in maintaining brain homeostasis in health and disease, synapse formation, neuronal proliferation, and brain development. Dysregulation of these processes can impact learning and memory, and deficits have been associated with increased and prolonged microglial activation after infection with SARS-CoV-2, but also with other viruses like influenza A viruses [43–45].

The role of astrocytic activation and Alzheimer type II astrocytes triggered by SARS-CoV-2 infection has been less studied. One study reported an increase in astrocytic activation [28], while another study reported no differences in a hamster model [24]. The functional consequences of decreased astrocyte activation are currently unknown, but altered astrocyte physiology can modulate both astrocytic and neuronal glutamate transporter trafficking and activity [46]. Thus, a reduction of the reactive astrocyte marker GFAP, meaning that the astrocytes are in a less reactive state, can play a role in dysregulation of glutamate signaling. Altered glutamate signaling is implicated in neuropsychiatric disorders and neurodegenerative diseases, which are both associated with SARS-CoV-2 infection [26,40,43,47,48]. Alzheimer type II astrocytes, which we detected in the hindbrain, have recently been detected in the brain of a COVID-19 infected patient. This patient was diagnosed with viral meningoencephalitis with symptoms like altered consciousness and seizures [25]. Alzheimer type II astrocytes may be an indicator of hepatic encephalopathy [35,49] but have not been studied extensively. The implications of the presence of Alzheimer type II astrocytes after SARS-CoV-2 infection are currently unknown and need to be studied more.

Limitations of this study are the limited sample size and the limited time points post inoculation. Despite these limitations, it would be interesting to follow-up whether the observed neurovirulent potential of SARS-CoV-2 would result in functional consequences. Therefore, future studies focused on the neurovirulent potential of SARS-CoV-2 should include behavioural studies, like the buried food test to study anosmia [50] or the Morris water maze, to study learning and memory processes. Additionally, electrophysiology could elucidate changes in synaptic plasticity upon infection, which is observed after infection with other respiratory viruses like influenza A viruses [44,51].

Together, these data suggest that the ferret serves as a model for self-limiting to mild respiratory disease with limited neuroinvasive potential. Even though the lack of evidence for neuroinvasion, the infection still induced neurovirulence evident by microglial activation, astrocytic deactivation and Alzheimer type II astrocytes. So far, the neuropathogenesis of SARS-CoV-2 infections has mainly been studied in animal models for severe disease, while this study highlights the value of this ferret model to study the effects on the brain after mild respiratory disease following intranasal SARS-CoV-2 infection in the acute and post-acute phase.

## Supporting information

**S1 Data. Raw experimental data Fig 2.**
(XLSX)

**S2 Data. Raw experimental data Fig 3.**
(XLSX)

**S3 Data. Raw experimental data Fig 4.**
(XLSX)

## Author contributions

**Conceptualization:** Feline F. W. Benavides, Lisa Bauer, Debby van Riel.

**Data curation:** Feline F. W. Benavides, Lonneke Leijten.

**Formal analysis:** Feline F. W. Benavides, Edwin J. B. Veldhuis Kroeze, Lonneke Leijten.

**Funding acquisition:** Rory D. de Vries, Debby van Riel.

**Investigation:** Feline F. W. Benavides, Edwin J. B. Veldhuis Kroeze, Lonneke Leijten, Peter van Run.

**Methodology:** Feline F. W. Benavides, Lonneke Leijten, Katharina S. Schmitz, Rory D. de Vries.

**Project administration:** Feline F. W. Benavides, Katharina S. Schmitz, Rory D. de Vries, Lisa Bauer, Debby van Riel.

**Resources:** Katharina S. Schmitz, Rory D. de Vries.

**Software:** Feline F. W. Benavides.

**Supervision:** Thijs Kuiken, Lisa Bauer, Debby van Riel.

**Validation:** Feline F. W. Benavides, Edwin J. B. Veldhuis Kroeze, Lonneke Leijten.

**Visualization:** Feline F. W. Benavides.

**Writing – original draft:** Feline F. W. Benavides, Thijs Kuiken, Lisa Bauer, Debby van Riel.

**Writing – review & editing:** Feline F. W. Benavides, Edwin J. B. Veldhuis Kroeze, Lonneke Leijten, Katharina S. Schmitz, Peter van Run, Thijs Kuiken, Rory D. de Vries, Lisa Bauer, Debby van Riel.

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
