## [Decision Letter · Decision Letter 0]

7 Jan 2025

PONE-D-24-41674Neuroinvasive and neurovirulent potential of SARS-CoV-2 in the acute and post-acute phase of intranasally inoculated ferretsPLOS ONE

Dear Dr. van Riel,

Thank you for submitting your manuscript to PLOS ONE. After careful consideration, we feel that it has merit but does not fully meet PLOS ONE’s publication criteria as it currently stands. Therefore, we invite you to submit a revised version of the manuscript that addresses the points raised during the review process.

 During the revision process, please clarify the position on the neuroinvasive conclusions drawn based on the data presented.  

We look forward to receiving your revised manuscript.

Kind regards,

Victor C Huber

Academic Editor

PLOS ONE

Journal Requirements:

This work was funded in part by a fellowship from the Netherlands Organization for Scientific Research (VIDI contract 91718308).  Ferret experiments were previously published and supported by funding from the National Institutes of Health (AI146980, AI121349, NS091263, AI114736, HHSN272201400008C).  

This work was funded in part by a fellowship from the Netherlands Organization for Scientific Research (VIDI contract 91718308). Ferret experiments were previously published and supported by funding from the National Institutes of Health (AI146980, AI121349, NS091263, AI114736, HHSN272201400008C).

This work was funded in part by a fellowship from the Netherlands Organization for Scientific Research (VIDI contract 91718308).  Ferret experiments were previously published and supported by funding from the National Institutes of Health (AI146980, AI121349, NS091263, AI114736, HHSN272201400008C). 

5. We note that your Data Availability Statement is currently as follows: All relevant data are within the manuscript and its Supporting Information files.

Reviewers' comments:

Reviewer's Responses to Questions

**Comments to the Author**

1. Is the manuscript technically sound, and do the data support the conclusions?

Reviewer #1: Partly

Reviewer #2: Yes

2. Has the statistical analysis been performed appropriately and rigorously? 

Reviewer #1: Yes

Reviewer #2: Yes

3. Have the authors made all data underlying the findings in their manuscript fully available?

Reviewer #1: Yes

Reviewer #2: Yes

4. Is the manuscript presented in an intelligible fashion and written in standard English?

Reviewer #1: Yes

Reviewer #2: Yes

5. Review Comments to the Author

Reviewer #1: The manuscript titled “Neuroinvasive and neurovirulent potential of SARS-CoV-2 in the acute and post-acute phase of intranasally inoculated ferrets” by Benavides et al. examines the potential of SARS-CoV-2 to invade the brain and cause neurological changes in a ferret model. The study offers valuable insights into how even mild respiratory infections might impact the brain, highlighting subtle changes in microglial and astrocytic activity. However, several methodological limitations need to be addressed to enhance the robustness of the findings.

Major Concerns:

1. While the study detects viral RNA in the brain through qPCR, this is not confirmed by in situ hybridization (ISH) or immunohistochemistry (IHC). This weakens the claim of neuroinvasion. Employing electron microscopy, at earlier time points (e.g., days 2–4) could provide stronger evidence.

2. Although changes in microglial and astrocytic activity are reported, the study does not explore functional consequences such as cognitive or behavioral changes. Including behavioral data or electrophysiological studies in future work would significantly enhance the interpretation of these findings. If these experiments are not feasible at this stage, the manuscript should include a discussion of potential follow-up studies to explore functional outcomes and current limitations.

3. The small sample size limits the statistical power and generalizability of the results. The authors should acknowledge this limitation and propose including larger cohorts in future research to address inter-individual variability.

4. Sampling at 7 and 21 dpi may overlook critical early events of neuroinvasion or inflammatory response. Including earlier time points (e.g., 1–4 dpi) in future studies is recommended. The authors should discuss this limitation in the current manuscript.

5. Incorporating systemic cytokine profiling would provide valuable information about potential indirect mechanisms driving neurovirulence, adding depth to the findings.

Minor Concerns:

1. While the figures and tables are informative, adding more detailed statistical annotations would improve clarity and accessibility for readers.

2. The extent of GFAP changes in areas beyond the olfactory bulb is unclear. This should be clarified.

Conclusion:

This study offers important insights into the neurological effects of SARS-CoV-2 in a mild respiratory disease model and underscores the relevance of the ferret model in this context. However, the absence of direct evidence for neuroinvasion, limited exploration of functional consequences, and other methodological constraints reduce its impact. Addressing these concerns, either experimentally or through a detailed discussion, would strengthen the manuscript and make its contributions clearer to the readership.

Recommendation: Revisions required.

Reviewer #2: The authors describe a study on the neuroinvasive and neurovirulent potential of SARS-CoV-2 conducted on the ferret animal model, specifically those that had subclinical or mild infection for which it is currently unknown. It was noted that tissues analysed in the present work were collected from a previous study (ref 29) on intranasally inoculated SARS-CoV-2 ferrets. Findings on the limited neuroinvasive potential of the virus warrants further work on the neurological effects of subclinical/mild disease.

Methods and results were generally well described however, the writing would benefit from editing/language checks with the following observed upon review:

Line 64: “In experimentally inoculation animals…” suggest to check the term used.

Line 91: "...infection is currently unknown, Therefore we investigated the neuroinvasive..."

Other comments:

Line 71: Authors described the mechanisms of neurovirulence, but in what way are microglia/astrocytes activated with SARS-CoV-2 infection? E.g. neuroinflammatory response, activation of mast cells?

Line 298: A number of references were listed i.e. (4,5,14,16,17,29) - it would be worth providing some description of other similar studies e.g. in ferrets, relating to the results of this study and how they differ/align.

6. PLOS authors have the option to publish the peer review history of their article (what does this mean? ). If published, this will include your full peer review and any attached files.

**Do you want your identity to be public for this peer review?** For information about this choice, including consent withdrawal, please see our Privacy Policy .

Reviewer #1: No

Reviewer #2: No

---

## [Author Response · Author response to Decision Letter 0]

4 Feb 2025

We thank the reviewers for their time to comment on this manuscript. Below we have addressed all their comments.

Reviewer #1: The manuscript titled “Neuroinvasive and neurovirulent potential of SARS-CoV-2 in the acute and post-acute phase of intranasally inoculated ferrets” by Benavides et al. examines the potential of SARS-CoV-2 to invade the brain and cause neurological changes in a ferret model. The study offers valuable insights into how even mild respiratory infections might impact the brain, highlighting subtle changes in microglial and astrocytic activity. However, several methodological limitations need to be addressed to enhance the robustness of the findings.

Major Concerns:

1. While the study detects viral RNA in the brain through qPCR, this is not confirmed by in situ hybridization (ISH) or immunohistochemistry (IHC). This weakens the claim of neuroinvasion. Employing electron microscopy, at earlier time points (e.g., days 2–4) could provide stronger evidence.

We agree that we cannot make any strong conclusions about neuroinvasion at earlier time points, however, the fact that we detect viral RNA in a few samples suggests that this might be the case. This is also supported by van de Ven et al., (2021) Frontiers of Immunology. This is included in the discussion (line 307-308). In future studies, sampling earlier time points for the detection of viral antigen or RNA, or virus particles by immunohistochemistry or in situ hybridization or EM respectively would indeed be favorable.

2. Although changes in microglial and astrocytic activity are reported, the study does not explore functional consequences such as cognitive or behavioral changes. Including behavioral data or electrophysiological studies in future work would significantly enhance the interpretation of these findings. If these experiments are not feasible at this stage, the manuscript should include a discussion of potential follow-up studies to explore functional outcomes and current limitations.

We agree that functional studies would provide important new insights although this might be challenging in ferrets as this has not been established. This is now discussed in lines 342-347.

3. The small sample size limits the statistical power and generalizability of the results. The authors should acknowledge this limitation and propose including larger cohorts in future research to address inter-individual variability.

We agree that the sample size is limited. This is now discussed in lines 341-343.

4. Sampling at 7 and 21 dpi may overlook critical early events of neuroinvasion or inflammatory response. Including earlier time points (e.g., 1–4 dpi) in future studies is recommended. The authors should discuss this limitation in the current manuscript.

As discussed with point 2 of this reviewer, this is now discussed.

5. Incorporating systemic cytokine profiling would provide valuable information about potential indirect mechanisms driving neurovirulence, adding depth to the findings.

We agree that systemic cytokines could trigger neurovirulence. This is discussed in line 315-319.

Minor Concerns:

1. While the figures and tables are informative, adding more detailed statistical annotations would improve clarity and accessibility for readers.

This information is now added in line 219-220.

2. The extent of GFAP changes in areas beyond the olfactory bulb is unclear. This should be clarified.

Changes in GFAP expression (and IBA1) at the level of the grey matter, white matter and hippocampus are depicted in figure 4. We discussed these data in the discussion, line 326-340.

Conclusion:

This study offers important insights into the neurological effects of SARS-CoV-2 in a mild respiratory disease model and underscores the relevance of the ferret model in this context. However, the absence of direct evidence for neuroinvasion, limited exploration of functional consequences, and other methodological constraints reduce its impact. Addressing these concerns, either experimentally or through a detailed discussion, would strengthen the manuscript and make its contributions clearer to the readership.

Recommendation: Revisions required.

Reviewer #2: The authors describe a study on the neuroinvasive and neurovirulent potential of SARS-CoV-2 conducted on the ferret animal model, specifically those that had subclinical or mild infection for which it is currently unknown. It was noted that tissues analysed in the present work were collected from a previous study (ref 29) on intranasally inoculated SARS-CoV-2 ferrets. Findings on the limited neuroinvasive potential of the virus warrants further work on the neurological effects of subclinical/mild disease.

Methods and results were generally well described however, the writing would benefit from editing/language checks with the following observed upon review:

Line 64: “In experimentally inoculation animals…” suggest to check the term used.

We adjusted this accordingly at line 64-65.

Line 91: "...infection is currently unknown, Therefore we investigated the neuroinvasive..."

We adjusted this accordingly at line 93.

Other comments:

Line 71: Authors described the mechanisms of neurovirulence, but in what way are microglia/astrocytes activated with SARS-CoV-2 infection? E.g. neuroinflammatory response, activation of mast cells?

We agree with the reviewer that the neurovirulence of SARS-CoV-2 is likely complex and multifactorial. The discussion has been changed at line 315.

Line 298: A number of references were listed i.e. (4,5,14,16,17,29) - it would be worth providing some description of other similar studies e.g. in ferrets, relating to the results of this study and how they differ/align.

We have now included all available (to our knowledge) literature on SARS-CoV-2 inoculated ferrets that included the neuroinvasive, neurotropic or neurovirulent potential of SARS-CoV-2.

---

## [Editor Report · Decision Letter 1]

13 Feb 2025

Neuroinvasive and neurovirulent potential of SARS-CoV-2 in the acute and post-acute phase of intranasally inoculated ferrets

PONE-D-24-41674R1

Dear Dr. van Riel,

We’re pleased to inform you that your manuscript has been judged scientifically suitable for publication and will be formally accepted for publication once it meets all outstanding technical requirements.

Kind regards,

Victor C Huber

Academic Editor

PLOS ONE
---

## [Editor Report · Acceptance letter]

PONE-D-24-41674R1

PLOS ONE

Dear Dr. van Riel,

I'm pleased to inform you that your manuscript has been deemed suitable for publication in PLOS ONE. Congratulations! Your manuscript is now being handed over to our production team.

Kind regards,

on behalf of

Dr. Victor C Huber

Academic Editor

PLOS ONE